# Royal Jelly Abrogates Cadmium-Induced Oxidative Challenge in Mouse Testes: Involvement of the Nrf2 Pathway

**DOI:** 10.3390/ijms19123979

**Published:** 2018-12-10

**Authors:** Rafa S. Almeer, Doaa Soliman, Rami B. Kassab, Gadah I. AlBasher, Saud Alarifi, Saad Alkahtani, Daoud Ali, Dina Metwally, Ahmed E. Abdel Moneim

**Affiliations:** 1Department of Zoology, College of Science, King Saud University, Riyadh 11495, Saudi Arabia; galbeshr@gmail.com (G.I.A.); salarifi@ksu.edu.sa (S.A.); salkahtani@KSU.EDU.SA (S.A.); daudali.ksu12@yahoo.com (D.A.); mdbody7@yahoo.com (D.M.); 2Department of Zoology and Entomology, Faculty of Science, Helwan University, Cairo 11795, Egypt; ganah_science@yahoo.com (D.S.); rami.kassap@yahoo.com (R.B.K.); aest1977@hotmail.com (A.E.A.M.); 3Parasitology Department, Faculty of Veterinary Medicine, Zagazig University, Zagazig 44519, Egypt

**Keywords:** royal jelly, cadmium, oxidative stress, mice, testes

## Abstract

The current study examined the efficacy of royal jelly (RJ) against cadmium chloride (CdCl_2_)-induced testicular dysfunction. A total of 28 Swiss male mice were allocated into four groups (n = 7), and are listed as follows: (1) the control group, who was intraperitoneally injected with physiological saline (0.9% NaCl) for 7 days; (2) the RJ group, who was orally supplemented with RJ (85 mg/kg daily equivalent to 250 mg crude RJ) for 7 days; (3) the CdCl_2_ group, who was intraperitoneally injected with 6.5 mg/kg for 7 days; and (4) the fourth group, who was supplemented with RJ 1 h before CdCl_2_ injection for 7 days. Cd-intoxicated mice exhibited a decrease in serum testosterone, luteinizing hormone (LH), and follicle stimulating hormone (FSH) levels. A disturbance in the redox status in the testicular tissue was recorded, as presented by the increase in lipid peroxidation and nitrate/nitrite levels and glutathione (GSH) depletion. Moreover, the activities of glutathione peroxidase (GPx), glutathione reductase (GR), superoxide dismutase (SOD), catalase (CAT), and nuclear factor (erythroid-derived 2)-like-2 factor (Nrf2) and their gene expression were inhibited. In addition, interleukin-1ß (IL-1β) and tumor necrosis factor-α (TNF-α) levels were elevated. Furthermore, Cd triggered an apoptotic cascade via upregulation of caspase-3 and Bax and downregulation of Bcl-2. Histopathological examination showed degenerative changes in spermatogenic cells, detachment of the spermatogenic epithelium from the basement membrane, and vacuolated seminiferous tubules. Decreased cell proliferation was reflected by a decrease in proliferating cell nuclear antigen (PCNA) expression. Interestingly, RJ supplementation markedly minimized the biochemical and molecular histopathological changes in testes tissue in response to Cd exposure. The beneficial effects of RJ could be attributed to its antioxidative properties.

## 1. Introduction 

Cadmium (Cd) is a heavy metal, and is placed between the most ubiquitous environmental toxicants that cause deleterious side effects to the living organisms at low or high levels [1]. Humans are mainly exposed to Cd through inhalation and ingestion from different sources, including contaminated water, food, and smoking, but also from several industrial products such as paints, batteries, fertilizers, and plastics [2]. Cd salts are characterized by great stability, low excretion rate, and long biological half-life, which may extend upwards of 20 years [3]. Once absorbed, Cd accumulates in various organs, especially brain, liver, bones, kidney, and testes [4]. 

Cd is associated with the development of several health problems, including reproductive dysfunction, renal failure, cancer, cardiovascular diseases, and neurological disorders [2]. Moreover, numerous studies have reported that Cd produces reproductive dysfunctions by disturbing the redox balance, blood-testis barrier, sex hormones homeostasis, sperm count, and enhancing germ cells lose [5,6,7]. Although the precise mechanisms of adverse reactions following Cd exposure are not fully explained, oxidative stress has been reported to play a fundamental role in Cd-induced tissue injuries [8]. Cd has been tightly linked with the overproduction of reactive oxygen species (ROS), which interact with proteins, lipids, carbohydrates, and DNA, and subsequently produce many pathological conditions [2]. In addition, Cd was found to potentiate apoptotic cascades and alter the ratio between the pro- and anti-apoptotic proteins [9]. Furthermore, Cd is able to enhance a massive cellular inflammatory status through the excessive release of pro-inflammatory mediators, namely interleukin-1 β (IL-1β), tumor necrosis factor-α (TNF-α), and nitric oxide (NO) (Elmallah et al. 2017). 

Treatment of Cd intoxication using antioxidants produced with the chemical chelating agents has been suggested as an accepted strategy, considering their efficiency and low side effects [1]. Natural products and their polyphenolic active constituents are well known to protect the cellular macromolecules by scavenging ROS and increasing the activity for cellular antioxidant molecules. For thousands of years, honey bees and their products have not only been used as a food source, but for the treatment of many disorders, due to their rich nutraceutical and pharmacological capacity [10]. 

Royal Jelly (RJ) is produced from the hypopharyngeal and mandibular glands of nurse bees (*Apis mellifera* L.) as a white viscous secretion with a pH between 3.6 and 4.2. RJ is used as a food that directs the development of larva into the queen bees [11]. RJ is a mixture of water, carbohydrates (mainly fructose, glucose, and sucrose), proteins (mainly the Major RJ Proteins [MRJPs]), fatty acids (mainly 10-hydroxy-2-decenoic [10-HDA]), vitamins (mainly vitamin B5, thiamin, niacin, and riboflavin), and mineral salts (mainly iron, zinc, and calcium) [12,13]. RJ has been associated with many biological and pharmacological functions, such as an anti-inflammatory [14], antioxidant [9], anti-tumor [15], anti-bacterial [14], anti-diabetic [16], anti-hypertensive [17], and immuno-modulator [18]. 

Our previous findings showed the protective efficiency of RJ supplementation in Cd-induced hepato- and neurotoxicity. Our interest has further risen to assess the potential protective role of RJ against Cd-induced testicular dysfunction in rats by evaluating the levels of sex hormones [testosterone, luteinizing hormone (LH), and follicle-stimulating hormone (FSH)], oxidants [lipid peroxidation (LPO) and nitrate/nitrite], antioxidants [glutathione (GSH), glutathione peroxidase (GPx), glutathione reductase (GR), superoxide dismutase (SOD), catalase (CAT), and nuclear factor (erythroid-derived 2)-like-2 factor (Nrf2) and their gene expression], pro-inflammatory cytokines [Interleukin-1ß (IL-1β) and Tumor Necrosis Factor-α (TNF-α)], apoptotic proteins [caspase-3, Bcl-2-associated X (Bax), and B-cell lymphoma 2 (Bcl-2)], and proliferating cell nuclear antigen (PCNA) expression.

## 2. Results 

### 2.1. Effect of RJ on Cd Accumulation in the Testicular Tissue

Cd concentration was observed to have increased significantly (*p* < 0.05) in testicular homogenates of mice injected intraperitoneally with CdCl_2_ for 7 days at a dose of 6.5 mg/kg when compared with the control group. The level of this heavy metal was non-significantly changed in the RJ treated group at a dose of 85 mg/kg. Moreover, RJ supplementation 1 h before CdCl_2_ treatment significantly attenuated Cd accumulation in testicular tissue compared to mice treated with CdCl_2_ alone (Figure 1).

### 2.2. Effect of RJ and Cd on the Absolute and Relative Weight of Testes 

The effect of Cd on testicle absolute and relative weight has been illustrated in Figure 2. In comparison to the control group, Cd-intoxicated mice also showed a significant decrease (*p* < 0.05) in the testicular absolute and relative weight. Interestingly, oral administration of RJ significantly increased the testicular relative weight, but not the absolute weight, as compared against the control mice. Furthermore, RJ pretreatment significantly suppressed the decrease in the testes weight caused by CdCl_2_ injection.

### 2.3. Effect of RJ and Cd on Serum Levels of Testosterone, LH, and FSH

In the current study, we assessed male sex hormone alteration following Cd exposure. Serological testosterone, LH, and FSH concentrations were found to be significantly declined (*p* < 0.05) in Cd-intoxicated mice when compared to their corresponding levels in control mice. In contrast, RJ supplemented mice exhibited an increase in the level of testosterone and FSH, suggesting that RJ may have a role in testosterone and FSH synthesis. RJ and CdCl_2_ treated mice showed a marked increase in testosterone, LH, and FSH concentration, compared to CdCl_2_-only treated mice. However, the hormonal level was still significantly lower than the control group (Figure 3). 

### 2.4. Effect of RJ on Cd-Induced Oxidative Stress in the Testicular Tissue

Mice exposed to CdCl_2_ showed a perturbation in the redox status in the testicular homogenates as indicated by the marked elevation (*p* < 0.05) of LP and NO. The increment of these oxidants was associated with a decrease in GSH content and the activities of GPx, GR, SOD, and CAT, as compared with their levels in the control mice. RJ-gavaged mice recorded a non-significant change in the levels of the aforementioned oxidative stress markers after 7 days. Meanwhile, RJ pretreatment restored the oxidant/antioxidant balance in the testicular homogenates of Cd-intoxicated mice (Figure 4 and Figure 5). Consistent with these results, RT-qPCR data showed that mRNA expression of GPx1, GR, SOD2, and CAT was significantly downregulated (*p* < 0.05) following Cd exposure as compared to their corresponding expression levels in control mice. No marked change in the expression of these antioxidant enzymes was recorded in the RJ orally treated group. Meanwhile, RJ treatment prior to CdCl_2_ clarified the ability of RJ to significantly increase the mRNA expression of these endogenous antioxidants in the testicular tissue as compared with the CdCl_2_-injected group. In order to understand the molecular antioxidant properties of RJ, Nrf2 expression—which enhances the expression of cellular antioxidants and detoxifies the system—has been estimated using RT-qPCR. In comparison to the control group, our findings revealed a significant downregulation (*p* < 0.05) of Nrf2 in CdCl_2_-exposed mice (Figure 6). Conversely, Nrf2 expression was upregulated significantly following RJ supplementation as compared to the control values. In addition, RJ also was recorded to normalize the Nrf2 expression in RJ and CdCl_2_ treated mice, reflecting its potent antioxidant activity against Cd-induced oxidative stress in rat testes.

### 2.5. Effect of RJ on Cd induced-Inflammatory Status in the Testicular Tissue

To evaluate the potential anti-inflammatory activity of RJ following CdCl_2_ intoxication, the levels of IL-1β and TNF-α were estimated in the testicular tissue. ELISA results showed a significant increase (*p* < 0.05) in IL-1β and TNF-α levels in Cd-exposed mice as compared with the control values. No change in the concentration of these chemical messengers was observed following RJ treatment. Meanwhile, IL-1β and TNF-α were decreased in the RJ and CdCl_2_ treated group (Figure 7).

### 2.6. Histopathological Changes in Testicular Tissue Following RJ and/or CdCl_2_

Histopathological investigation with light microscopy (Figure 7) of the testes of the control and RJ-treated groups (Figure 8a,b, respectively) exhibited a typical testicular histology with normal and functional seminiferous tubules with all stages of the spermatogenic cells and the Leydig cells filling the space between the seminiferous tubules. The testes from Cd-treated animals displayed many histopathological alterations, including degenerative changes in spermatogenic cells, detachment of the spermatogenic epithelium from the basement membrane, and appearance of vacuolated area in the seminiferous tubules (Figure 8c), whereas RJ-pre-administration significantly alleviated these abnormalities (Figure 8d). 

### 2.7. Effect of RJ on Cd-Triggered Apoptosis and Cytotoxicity in the Testicular Tissue

Immunohistochemical analysis clearly showed that CdCl_2_ potentiates the apoptotic cascade in the testicular tissue as presented by increasing the immunostaining intensity signal for the anti-apoptotic proteins, including caspase-3 and Bax, and decreasing the immunoreactivity of the anti-apoptotic, Bcl-2. RJ exhibited no significant immunoreactivity to the tested apoptotic markers. Meanwhile, supplementation of RJ before CdCl_2_ significantly decreased the number of the positively stained spermatogenic cells for caspase-3 and Bax, as well as increased Bcl-2 immunoreactivity. Moreover, the immunohistochemical examination displayed a marked depletion in PCNA immunostained cells in germinal cells of the Cd-injected group. Furthermore, mice treated with RJ and CdCl_2_ showed an increase in PCNA immunostaining intensity as compared to its corresponding expression in CdCl_2_-exposed mice (Figure 9). 

## 3. Discussion

Environmental and occupational exposure to heavy metals, including Cd, has been tightly associated with the progression of several pathological impairments by creating oxidative reactions. Testes have been classified among the most negatively affected organs following Cd intoxication. Here, we tried to investigate the potential protective efficiency of RJ against CdCl_2_-induced testicular dysfunction in mice. 

Organ weight has been widely used as an important toxicological marker [19]. Our results recorded a decrease in the absolute and relative weight of the testes after 7 days of treatment with CdCl_2_. This weight decrease may be due to a decrease in food intake, which has been reported to decrease germ cell numbers that cause testicular dysfunction [20,21]. In the same context, the decreased testes weight was accompanied by a decrease in the levels of serological testosterone, LH, and FSH in response to Cd exposure. According to previous studies, the decrease in the assessed sex hormones is due to the inactivation of steroidogenic enzymes, including 3β-and 17β-hydroxysteroid dehydrogenase, which disturbs the synthesis of androgen and suppresses testosterone production [22]. Additionally, Cd impairs the hormonal receptors and affecting hypothalamic-pituitary-gonadal axis, which inhibit steroidogenesis and further spermatogenesis [23]. Meanwhile, RJ pretreatment was able to protect and minimize Cd-induced testicular weight loss; this may be due to its high and rich nutraceutical constituents. RJ protection against Cd toxicity is extended to restore the levels of testosterone, LH, and FSH to be close to the normal values. 

Morita et al. [24] explained the increase of testosterone in volunteers who ingested RJ for 6 months as the ability of RJ to convert dehydroepiandrosterone sulfate (DHEA-S) into testosterone through androstenedione by 3β-and 17β-hydroxysteroid dehydrogenase found in the adrenal and testes; this effect could be due to RJ antioxidant enzyme activity. Moreover, 10-hydroxy-2-decenoic acid and MRJPs, the most active ingredients, in RJ were shown to have estrogenic-like effects in different animal models [25,26]. The increased testosterone concentration might be due to the presence of zinc in RJ. Zinc has been found to play a crucial role in spermatogenesis and its deficiency is associated with a decrease in testosterone concentration [27]. Furthermore, the amino acid content in RJ was found to have a role in androgen homeostasis [28]. An increase in Cd concentration was observed in the current study, suggesting its slow metabolism and excretion. Previous studies revealed that Cd exposure is associated with renal dysfunction, which further affects Cd elimination rate [29]. On the other hand, RJ was found to decrease Cd accumulation in the testicular homogenate. This behavior may be due to the metal chelating properties of the polyphenolic ingredients in RJ [30].

The obtained data further confirmed the involvement of oxidative challenge in Cd-induced testicular dysfunction, as indicated by the elevation of LPO and nitrate/nitrite levels and the deactivation of the cellular antioxidant and detoxifying molecules, including GSH, GPx, GR, SOD, CAT, and Nrf2 at the biochemical and molecular levels. Cd exposition has been linked to oxidant/antioxidant imbalance and ROS production in different body tissues. The produced free radicals enhance DNA oxidation, lipid peroxidation, and protein degradation leading to severe cytotoxic effects [31]. It is well known that Cd triggers fatty acids oxidation as a consequence of nitric oxide, hydroxyl radicals, superoxide anions, and hydrogen peroxide formation [32]. According to our previous study, the increased nitrate/nitrite level is due to iNOS upregulation which is responsible for nitric oxide production [9]. In addition, Cd interacts with sulfhydryl groups leading to GSH pool depletion. Meanwhile, our findings showed that the deactivation of endogenous antioxidant enzymes is due to the inhibition of their gene transcription in the testicular tissue. The suppression of GPx after Cd exposure has been attributed to the depletion of selenium, which plays a crucial role in GPx function [33]. Furthermore, the decrease in SOD could be due to the interaction between Cd and SOD which leads to changes in the enzyme structure, thus affecting its catalytic activity [34]. 

Jurczuk et al. [35] attributed CAT deactivation to the Fe deficiency, which is a necessary element in its active site. It has been suggested that the antioxidant agents may mitigate Cd-induced oxidative challenges due to their ability to scavenge free radicals and increase the activity of endogenous antioxidant defense systems [1]. Interestingly, RJ, in the current study, showed potent antioxidant activity against Cd-induced oxidative stress in the testicular tissue. RJ peptide content showed strong antioxidant properties and provided cellular protection through scavenging hydroxyl radicals and inhibiting lipid peroxidation in vivo and in vitro [36]. RJ supplementation downregulated the gene expression of the cytochrome P450 4A14 (CYP4A14) enzymes, which is responsible for the production of intracellular free radicals and inhibition enzymes which activate lipid peroxidation. Additionally, it upregulated gene expression of glutathione S-transferase and glutathione peroxidase in fumonisin-treated rats [37]. RJ was found to decrease malondialdehyde levels and increase the activity of GPx, SOD, and CAT in rats treated with cisplatin [38]. The antioxidant activity of RJ has been attributed to its free amino acids content [39]. Furthermore, RJ enhanced the activities of SOD and GPx and decreased malondialdehyde concentration in female diabetic rats [40]. 

In the present study, mice exposed to Cd showed a marked decrease in Nrf2 expression in the testicular tissue. Nrf2 is known to provide cellular protection against ROS via different mechanisms, including increasing GSH synthesis, upregulating antioxidant and detoxifying enzymes, and degrading superoxide and peroxide radicals by GPx and SOD [41]. In contrast, RJ administered mice minimize the Cd-induced oxidative stress through the upregulation of Nrf2. In our recent work, Nrf2 expression was also found to be upregulated in hepatic tissue of RJ treated mice following Cd intoxication [9]. Moreover, RJ protected lymphocytes against doxorubicin-induced oxidative stress by enhancing Nrf2 and homoxygenase-1 [42]. Additionally, 4-hydroperoxy-2-decenoic acid ethyl ester (HPO-DAEE), a lipid component in RJ, enhanced Nrf2 translocation to the nucleus, activating antioxidant elements, which in turn enhanced the gene expression of the antioxidant and detoxifying enzymes in 6-hydroxydopamine-induced cell death [43].

An inflammatory response in the testicular homogenate was recorded in our experiment following Cd-intoxication, as indicated by the increase of IL-1β and TNF-α. The overproduction of TNF-α in response to pathogens stimulates the release of other pro-inflammatory cytokines, including IL-β, which enhances gene expression associated with inflammation [44]. According to previous studies, this increase is due to ROS production, which activates NF-κB (nuclear factor-kappa B) translocation then upregulates IL-1β and TNF-α mRNA expression [45,46]. RJ treatment reversed the alterations of these pro-inflammatory cytokines. Several reports recorded the anti-inflammatory activity of RJ and its ingredients in different experimental models. RJ supplementation for 14 days at a dose of 300 mg/kg elicited a decrease in TNF-α levels in cyclophosphamide-treated rats [47]. Similarly, rats pre-administered with RJ (50 and 100 mg/kg/7 day) showed a marked decline in the concentration of TNF-α in response to a single dose of bleomycin [48]. Moreover, 10-hydroxy-2-decenoic acid, the major lipid constituent in RJ, was found to exert anti-inflammatory effects in human colon cancer cells via inhibiting NF-κB, which further suppressed the release of IL-1β and TNF-α [14].

In the current study, immunohistochemical investigation showed that Cd is potentiating apoptotic effect in the testes tissue by upregulating the pro-apoptotic proteins (caspase-3 and Bax) and downregulating Bcl-2, the anti-apoptotic protein in the testicular tissue. It has been proposed that Cd may mediate germ cell apoptosis with oxidative stress [49]. Oxidative stress is known to impair calcium ion channels and alter the mitochondrial membrane potential, leading to cytochrome C release, which enhances caspase cascade and fragmentation of DNA. RJ counteracted the apoptotic cascade produced by Cd [49]. On the other hand, RJ supplementation arrested the apoptotic progression and protected the testicular tissue. Rafat et al. [50] demonstrated that RJ consumption protected the human peripheral blood leukocytes against radiation-induced apoptosis. Moreover, RJ inhibited pro-apoptotic (caspase-3) and enhanced expression of the anti-apoptotic (Bcl-2) related proteins in liver and kidney tissues of cisplatin treated rats [51]. Bax, the pro-apoptotic protein, was found to be downregulated after RJ treatment in rats treated with cyclophosphamide [47]. The anti-apoptotic effect of RJ might be due to its antioxidant capacity [52].

PCNA is known to regulate cell cycle and DNA replication. In testes, PCNA has been used as a proliferative parameter to evaluate the spermatogenesis status. Cd-treated mice showed few positive immunostained germ cells for PCNA; this effect is suggested to be due to free radicals production, which enhances oxidation of DNA in spermatogenic cells [6]. Interestingly, RJ upregulated the expression of PCNA in the spermatogenic cells, which may be in part due to its antioxidant activity.

## 4. Materials and Methods

### 4.1. Chemicals

CdCl_2_ was supplied by Sigma Chemical Co. (St. Louis, MO, USA), whereas capsulated pure royal jelly was purchased from Pharco pharmaceuticals Co. Egypt and contains 6% 10-HDA. All other chemicals used for the experiments were of analytical grade.

### 4.2. Animals and Experimental Design

In total, 28 adult Swiss male mice, weighing 22–27 g, were attained from the Egyptian Organization for Biological Products and Vaccine. Each cage housed seven mice; the animals were given free access to water and a commercial pelleted rodent feed ad libitum. The mice were kept in the animal facility of the Zoology Department at Helwan University, Cairo, Egypt under standard conditions of laboratory with a temperature of 22–25 °C and a 12 h artificial light/dark cycle. 

The animals were treated per criteria of the Investigations and Ethics for Laboratory Animal Care at Zoology department, Faculty of Science, Helwan University (approval no, HU2017/Z/03). To investigate the prophylactic effect of RJ on Cd-induced oxidative damage in mice’s testes, the mice were randomly distributed into four equal groups (n = 7, each) after one week of acclimatization. Group I was the control; the animals in this group were intraperitoneally (i.p.) injected with physiological saline (0.9% NaCl) daily for 7 days. Group II were administered with RJ (85 mg/kg daily equivalent to 250 mg crude RJ dissolved in saline) for 7 days by oral gavage. Group III were i.p. treated with 6.5 mg/kg CdCl_2_ dissolved in physiological saline daily for 7 days. Group IV was supplemented with an oral administration of 85 mg/kg RJ 1 h before the i.p. injection of 6.5 mg/kg CdCl_2_ daily for 7 days. After 24 h of the last treatment, the mice were sacrificed using ether, then decapitated; the testes were rapidly excised, weighed, and then half of them directly homogenized in ice cold 10 mM phosphate buffer (pH 7.4) to prepare a 10% (*w*/*v*) homogenate, which was centrifuged at 4 °C for 10 min (10,000 rpm). The supernatant was obtained for further analyses, and the other half was fixed immediately for the histopathological and immunohistochemical examinations. For the determination of plasma testosterone, LH, and FSH, blood samples were also collected.

### 4.3. Estimation of Cadmium Concentration in Testes

Cd concentration in testes was measured by atomic absorption spectrophotometer (Perkin Elmer 3100), as previously described [53]. Briefly, an appropriate weight of testes was digested with 3 mL of nitric acid (HNO_3_) in a tightly capped 30 mL acid-washed polyethylene bottle. This bottle was left for 30 min at room temperature followed by an incubation at 70 °C in a water bath for 3 h. Sample volume was completed into 5 mL with HNO_3_ and then cured with an equal volume of 10% *v*/*v* H_2_O_2_. Finally, the solutions were incubated at room temperature for 10 min. Then, an appropriate sample volume was injected into a graphite furnace at 228 nm. Samples were analyzed in duplicate and Cd values were calculated from the standard curve on wet testes tissue basis in µg/g. 

### 4.4. Testicular Index

The testicular weight was measured using a sensitive weighing balance (Radwag, Model AS220/C/2, Clarkson laboratory and supply Inc., Chula Vista, CA, USA), whereas the relative testicular weight was calculated using the following formula:$$Relative\;Testicular\;Weight\;\left(RTW\right)=\frac{Left\;Testis\;\left(LT\right)}{Body\;weight}\times 100$$

### 4.5. Biochemical Analyses

#### 4.5.1. Hormones Measurements

Plasma FSH and LH were measured by double antibody radioimmunoassay, as previously described. Plasma testosterone was assayed according to the instructions in the kit’s manual (Elecsys Testosterone Assay Kits, Roche Diagnostics, Mannheim, Germany), by a microplate reader (Chromate awareness 4300, Palm City, FL, USA). Samples were calculated at 620 nm. 

#### 4.5.2. Lipid Peroxidation (LPO) 

The LPO index in 10% testicular homogenate was accomplished by estimating the concentration of malondialdehyde (MDA) according to the method described by Ohkawa et al. [54]. The developed color was determined as thiobarbituric acid reactive substances (TBARS) at 532 nm excitation and 555 nm emission. Accordingly, 100 mg of the testicular homogenate in phosphate buffer (pH 7.4) was mixed with 100 µL 100% trichloroacetic acid (TCA), 100 µL of sodium thioglycolate (1%), and 250 µL of 1 N HCl. The mixture was incubated for 20 min at 100 °C, and then centrifuged for 10 min (4000 rpm). Spectrophotometric determination of TBA-MDA complex was measured in the supernatant. 

#### 4.5.3. Nitrate/Nitrite Level

Valuation of NO in testicular homogenate was carried out according to the method of Sastry et al. through quantification of its stable products represented in both nitrite (NO_2_) nitrate (NO_3_) levels [55]. Briefly, 100 µL of the homogenate was added to 400 µL carbonate buffer and a trace amount of activated copper-cadmium alloy. The mixture was incubated at room temperature with continuous shaking. To stop the reaction, the alloy was removed and 100 µL of 0.35 M NaOH and 120 Mm ZnSO_4_ was added. The mixture was exposed to vigorous vortex and then permitted to stand for 10 min. Afterward, the mixture was centrifuged at room temperature for 10 min (4000 rpm). Griess reagent (50 µL) was added to 10 µL of the supernatant, incubated for 10 min, and lastly, the absorbance was determined at 545 nm using a microplate ELISA reader. 

#### 4.5.4. Reduced Glutathione (GSH) 

Testicular glutathione content in the testicular homogenate was detected by the method of Sedlak and Lindsay [56]. As such, 250 µL of 10% homogenate was added to 250 µL distilled water and 50 µL of 50% TCA. The mixture was exposed to successive shaking intervals for 15 min, then centrifuged at room temperature at 3000 rpm for 10 min. The supernatant (10 µL) was mixed with 400 µL 0.4 M Tris buffer (pH 8.9) and 10 µL of 5,5-dithio-bis-2-nitrobenzoic acid (DTNB) with continuous shaking. The color developed was evaluated at 512 nm by UV-VIS spectrophotometer (V-630; Jasco, Japan).

### 4.6. Catalase (CAT) Activity 

The activity of CAT was mainly determined based on the degradation rate of H_2_O_2_ per minute. The activity unit of CAT was showed as U/mg protein [57]. The total volume (1 mL) of the enzymatic reaction mixture was mainly consisted of 50 mM potassium phosphate (pH 7.0), 19 mM H_2_O_2_, and 50 µL of homogenate supernatant. The molar extinction coefficient of H_2_O_2_ was examined by UV-VIS spectrophotometer at 240 nm. One unit of enzyme activity was known as the amount of H_2_O_2_ (µmol) consumed per min per milligram of tissue protein (U/mg protein).

### 4.7. Superoxide Dismutase (SOD) Activity 

Testicular SOD activity was determined based on the method described by Misra and Fridovich [58]. The method depends on the susceptibility of epinephrine toward oxidation at higher pH (10.2) and generation of adrenochrome and superoxide radicals (O_2_^−^). The testicular SOD activity is then calculated by the degree of inhibition of this reaction by decreasing the absorbance at 480 nm. 

### 4.8. Glutathione Peroxidase (GPx) Activity

The activity of GPx in testicular homogenate was examined as previously described [59]. The reaction volume was adjusted to 1.5 mL. Briefly, 200 µL of testicular supernatant was mixed with 1 mL of 75 mM phosphate buffer (pH 7.0), 10 mL of 150 mM glutathione, 10 mL of 340 U/mL glutathione reductase, 30 mL of 25 mM EDTA, 30 mL of 5 mM NADPH, 10 mL of 20% Triton X-100, and 50 µL of 7.5 mM H_2_O_2_. The oxidation of GSH is linked to the conversion of NADPH (extinction coefficient = 6.22 3 × 10^3^ M^−1^ cm^−1^) to NADP^+^ that monitored at 340 nm for 3 min. One unit of GPx activity was known as the amount of GSH (nanomoles) oxidized per minute per milligram of protein (U/mg protein). 

### 4.9. Glutathione Reductase (GR) Activity 

The activity of GR enzyme was carried out by mixing the testicular supernatant (20 µL) with 0.44 mM oxidized glutathione (GSSG), 0.30 M EDTA, in 0.1 M phosphate buffer (pH 7.0). To begin the enzymatic reaction, 0.036 M NADPH was added. The rate of NADPH oxidation was observed by decreasing the absorbance at 340 nm as a function of time. One unit of enzyme was known as the amount of enzyme required to oxidize 1 µmol of NADPH per minute [60].

### 4.10. Determination of Testicular Levels of IL-1ß and TNF-α Levels

Quantitative measurements of IL-1ß (IL-1β; Cat. no. EM2IL1B, ThermoFisher Scientific, Waltham, MA, USA) and TNF-α (TNF-α; Cat. no. EZMTNFA, Millipore) levels were performed using enzyme-linked immunosorbent assay (ELISA) kits specified for mice according to the protocol provided with each kit.

### 4.11. Real Time-PCR

Isolation of total RNA from testes tissue was accomplished with a Trizol reagent and then converted to complementary DNA (cDNA) using cDNA Synthesis Kit (Bio-Rad, Hercules, CA, USA), according to the manufacturer’s instructions. For gene expression analysis, cDNA of the oxidative stress enzyme markers (CAT, SOD, GPx, and GR) were used as a template for quantitative Real-Time PCR. QuantiFast SYBR Green RT-PCR kit (Qiagen, Hilden, Germany) and the corresponding forward and reverse primers shown in Table 1 were implemented. Primers were acquired from (Jena Bioscience GmbH, Jena, Germany). All reactions were accomplished in triplicate using Applied Biosystems 7500 Instrument (ThermoFisher Scientific, CA, USA). The PCR cycling thermal conditions were established as follows: preliminary denaturation at 95 °C for 12 min, then by 40 cycles of denaturation at 94 °C for 60 s and annealing at 55 °C for 60 s, extension at 72 °C for 90 s, and afterwards held for a final extension at 72 °C for 10 min. The relative differences in gene expression between different groups were measured by delta-delta cycle threshold (Ct) method [61]. Glyceraldehyde-3-phosphate dehydrogenase (GAPDH) was utilized as a reference housekeeping gene. 

### 4.12. Histopathological Investigation 

The testes were removed from the sacrificed animals and fixed for 24 h at room temperature in 10% neutral-buffered formalin. The tissues were dehydrated in ascending series of alcohol, cleared in xylene, embedded in paraffin wax, and then sectioned at 5-μm thickness. The paraffin sections were washed with water, stained with hematoxylin and eosin [62], and examined using a light microscope. Images were obtained at an original magnification of 400× (Nikon Eclipse E200-LED, Tokyo, Japan).

### 4.13. Immunohistochemical Investigations

For immunohistochemistry investigations, the paraffin sections were mounted on glass slides and dewaxed. The antigen sites were revealed by washing the sections with boiled water after treatment for 10 min with 0.03% H_2_O_2_ in absolute methanol to stop endogenous peroxidase activity. Sections were incubated at 4 °C overnight with (1:50) polyclonal rabbit anti- Bcl-2 antibody, anti-Bax antibody, anti-caspases-3 antibody and anti-PCNA antibody (Santa Cruz, CA, USA). To get rid of the unbound primary antibodies, sections were washed with phosphate buffer saline (PBS). Afterward, sections were incubated for 30 min with goat-derived secondary anti-rabbit antibody conjugated to horseradish peroxidase at 37 °C. Antigen-antibody interactions were finally detected by incubating the sections for 10 min at room temperature with the chromogen 3,3′-diaminobenzidine tetrachloride (DAB-H2O2) as substrate. Testicular sections were visualized using 400× magnification lens (Nikon Eclipse E200-LED, Tokyo, Japan). 

### 4.14. Statistical Analysis 

Results are presented as the mean ± standard error of the mean values (SEM) of seven mice. Data from various evaluations were analyzed by one-way analysis of variance (ANOVA). Post hoc Duncan multiple tests were done. *p* values < 0.05 indicated statistical significance.

## 5. Conclusions

Overall, RJ supplementation was found to provide protection against Cd-induced testicular dysfunction. RJ ameliorated the hormonal alterations, oxidative status, inflammatory response, and the apoptotic cascade produced following Cd-exposure. These effects may be due to its potent antioxidant activity.

## Figures and Tables

**Figure 1 ijms-19-03979-f001:**
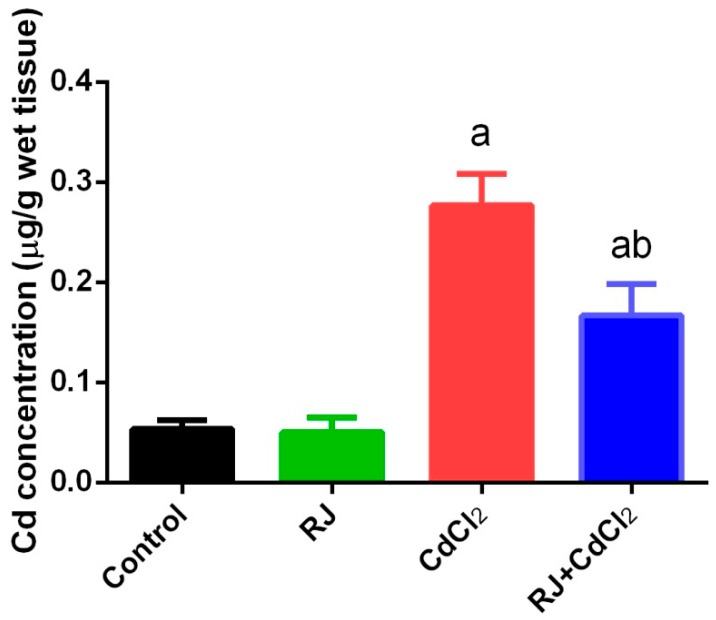
Effects of Royal jelly (RJ) administration on concentration of cadmium in testes of mice treated with cadmium chloride (CdCl_2_). All values are expressed as mean ± SEM (n = 7). ^a^ refers to a significant change from the control mice at *p* < 0.05; ^b^ refers to a significant change from the CdCl_2_-treated mice at *p* < 0.05, using Duncan’s post hoc test.

**Figure 2 ijms-19-03979-f002:**
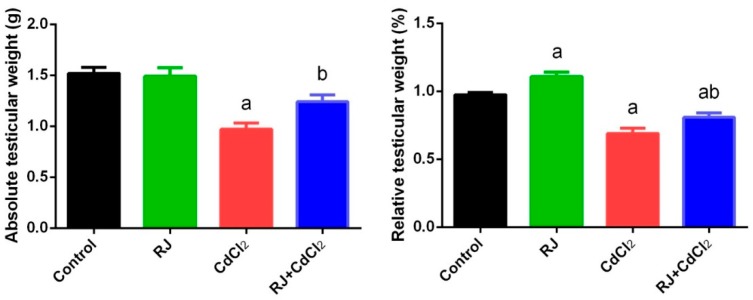
Effects of Royal jelly (RJ) administration on absolute and relative testicular weight of mice treated with cadmium chloride (CdCl_2_) toxicity. All values are expressed as mean ± SEM (n = 7). ^a^ refers to a significant change from the control mice at *p* < 0.05; ^b^ refers to a significant change from the CdCl_2_-treated mice at *p* < 0.05, using Duncan’s post hoc test.

**Figure 3 ijms-19-03979-f003:**
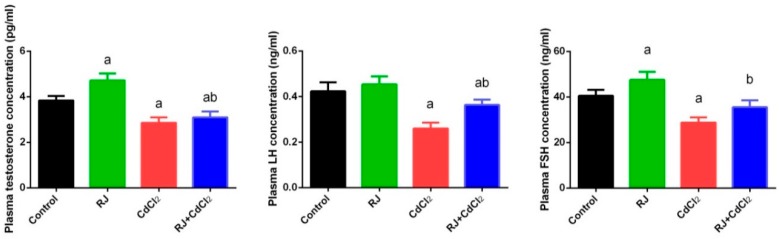
Effects of royal jelly (RJ) on serological testosterone, LH, and FSH levels of mice exposed to cadmium chloride (CdCl_2_). All values are expressed as mean ± SEM (n = 7). ^a^ refers to a significant change from the control mice at *p* < 0.05; ^b^ refers to a significant change from the CdCl_2_-treated mice at *p* < 0.05, using Duncan’s post hoc test.

**Figure 4 ijms-19-03979-f004:**
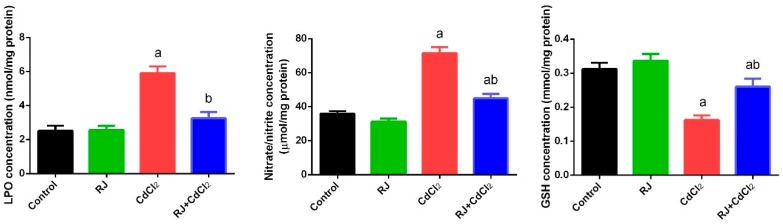
Effects of royal jelly (RJ) on lipid peroxidation (LPO), nitrate/nitrite, and reduced glutathione (GSH) levels in testes of mice exposed to cadmium chloride (CdCl_2_). All values are expressed as mean ± SEM (n = 7). ^a^ refers to a significant change from the control mice at *p* < 0.05; ^b^ refers to a significant change from the CdCl_2_-treated mice at *p* < 0.05, using Duncan’s post hoc test.

**Figure 5 ijms-19-03979-f005:**
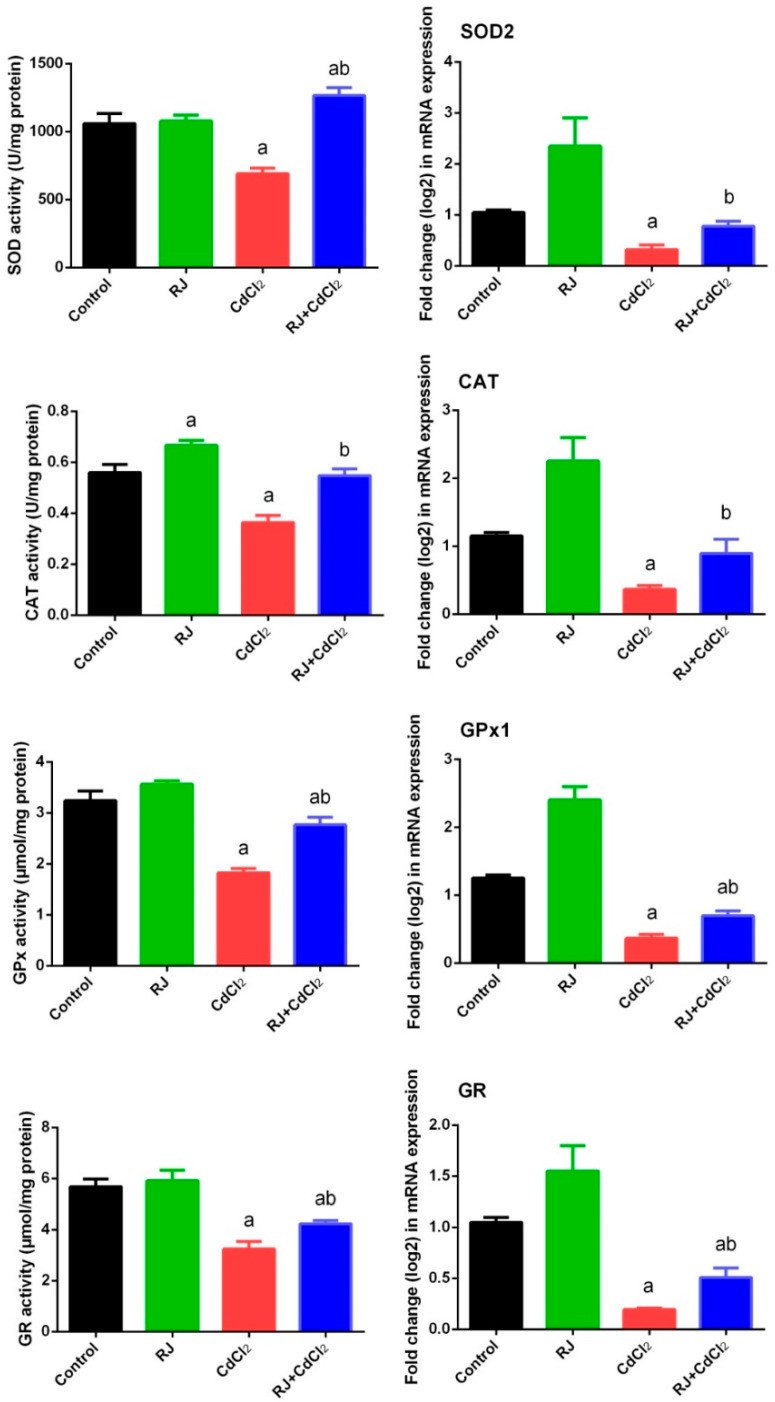
Effects of royal jelly (RJ) on the levels of superoxide dismutase (SOD), catalase (CAT), glutathione peroxidase (GPx), glutathione reductase (GR), and their corresponding mRNA expression in testes of mice exposed to cadmium chloride (CdCl_2_). Values of antioxidant enzyme activities are expressed as the mean ± SEM (n = 7), whereas data on mRNA expression levels (mean ± SEM of triplicate assays) were normalized to the GAPDH mRNA levels and are shown as the fold induction (in log2 scale) relative to the mRNA levels in the controls. ^a^ refers to a significant change from the control mice at *p* < 0.05; ^b^ refers to a significant change from the CdCl_2_-treated mice at *p* < 0.05, using Duncan’s post hoc test.

**Figure 6 ijms-19-03979-f006:**
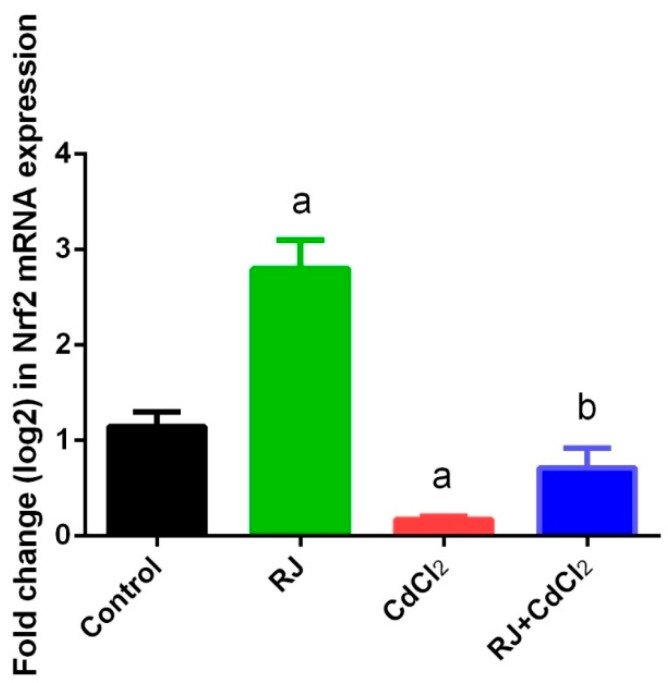
Effects of royal jelly (RJ) on the mRNA level of nuclear factor (erythroid-derived 2)-like-2 factor (Nrf2) in testes of mice exposed to cadmium chloride (CdCl_2_). All values are expressed as mean ± SEM of triplicate assays and were normalized to the GAPDH mRNA levels and are shown as the fold induction (in log2 scale) relative to the mRNA levels in the controls. ^a^ refers to a significant change from the control mice at *p* < 0.05; ^b^ refers to a significant change from the CdCl_2_-treated mice at *p* < 0.05, using Duncan’s post hoc test.

**Figure 7 ijms-19-03979-f007:**
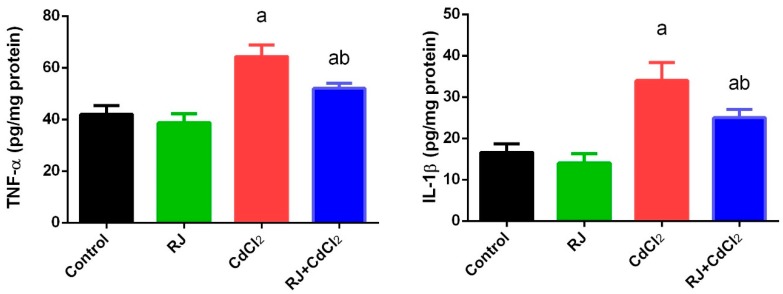
Effects of royal jelly (RJ) on the levels of tumor necrosis factor-α (TNF-α) and interleukin 1β (IL-1β) in testes of mice exposed to cadmium chloride (CdCl_2_). All values are expressed as mean ± SEM (n = 7). ^a^ refers to a significant change from the control mice at *p* < 0.05; ^b^ refers to a significant change from the CdCl_2_-treated mice at *p* < 0.05, using Duncan’s post hoc test.

**Figure 8 ijms-19-03979-f008:**
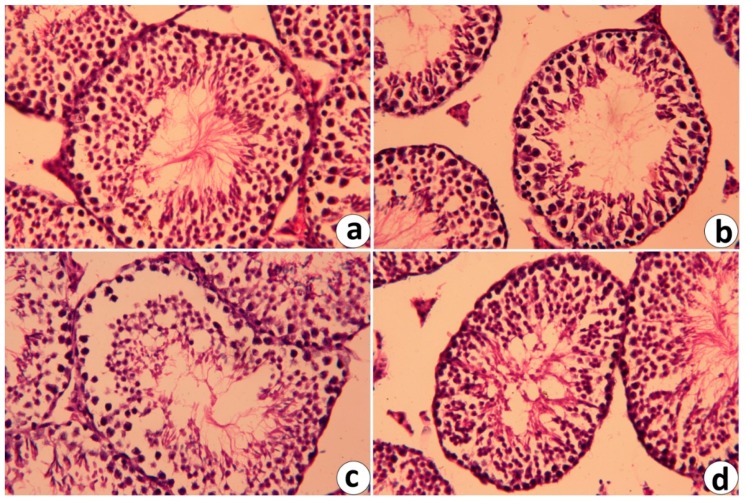
Photomicrographs of light microscope of testes of mice treated with RJ and CdCl_2_ for 7 days. Cross sections of testes were stained with hematoxylin and eosin (400×). (**a**,**b**) testes from control and royal jelly (RJ) groups, respectively exhibiting typical features of seminiferous epithelium and Leydig cells (LC). (**c**) testes from the CdCl_2_-treated mice showing degenerative seminiferous tubules. (**d**) testis from the pretreated group with RJ against CdCl_2_ showing an obvious preservation of spermatogenic epithelium in most seminiferous tubules.

**Figure 9 ijms-19-03979-f009:**
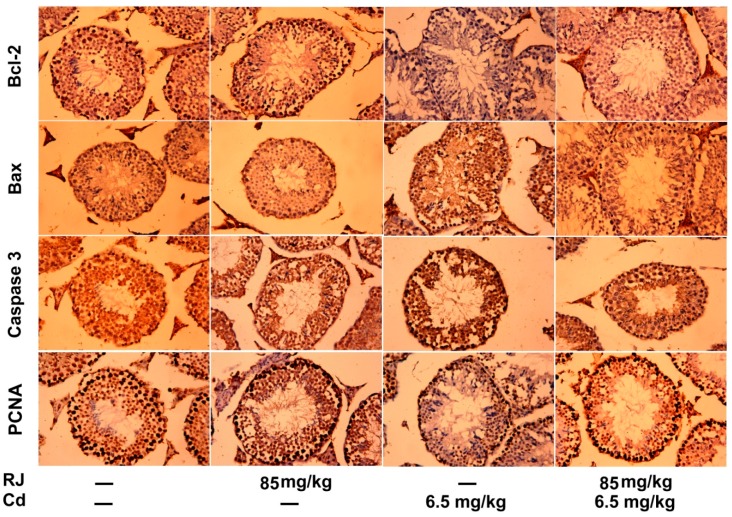
Photomicrographs showing changes in Bcl-2, Bax, caspase-3, and PCNA expression in the testicular tissue of mice following treatment with royal jelly (RJ) and cadmium chloride (CdCl_2_). (magnification, 400×).

**Table 1 ijms-19-03979-t001:** Primer sequences of genes analyzed in real time PCR.

Name	Accession Number	Forward Primer (5’---3’)	Reverse Primer (5’---3’)
**GAPDH**	NM_001289726.1	TCACCACCATGGAGAAGGC	GCTAAGCAGTTGGTGGTGCA
**SOD2**	NM_013671.3	GCCCAAACCTATCGTGTCCA	AGGGAACCCTAAATGCTGCC
**CAT**	NM_009804.2	CCGACCAGGGCATCAAAA	GAGGCCATAATCCGGATCTTC
**GSH-Px1**	NM_001329527.1	CAGCCGGAAAGAAAGCGATG	TTGCCATTCTGGTGTCCGAA
**GSH-R**	NM_010344.4	TGGCACTTGCGTGAATGTTG	CGAATGTTGCATAGCCGTGG
**Nrf2**	NM_010902.4	CCTCTGTCACCAGCTCAAGG	TTCTGGGCGGCGACTTTATT
**iNOS**	NM_001313922.1	CGAAACGCTTCACTTCCAA	TGAGCCTATATTGCTGTGGCT
**IL-1β**	NM_008361.4	TGCCACCTTTTGACAGTGATG	TTCTTGTGACCCTGAGCGAC
**TNF-α**	NM_013693.3	AGAGGCACTCCCCCAAAAGA	CGATCACCCCGAAGTTCAGT

The abbreviations of the genes are as follows: GAPDH, glyceraldehyde-3-phosphate dehydrogenase; SOD2, superoxide dismutase 2 mitochondrial (MnSOD); CAT, catalase; GSH-Px1, glutathione peroxidase 1; GSH-R, glutathione reductase; Nrf2, nuclear factor erythroid 2-related factor 2; iNOS, inducible nitric oxide synthase; IL-1β: Interleukin 1 beta; TNF-α: Tumor necrosis factor.

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
