# Peer review of "Royal Jelly Abrogates Cadmium-Induced Oxidative Challenge in Mouse Testes: Involvement of the Nrf2 Pathway"

_ijms, 2018, doi:10.3390/ijms19123979_

Round 1
Reviewer 1 Report
Comments to the Author;
This paper describes that royal jelly (RJ) administration ameliorated acute Cd-induced testicular dysfunction. RJ treatment mitigated Cd accumulation, serum hormone decline, and altered oxidative as well as inflammatory markers. RJ also normalized levels of antioxidant enzymes via an Nrf2-mediated antioxidant pathway. In addition, RJ treatment also alleviated histopathological changes and cellular toxicity in Cd-treated testis. This study would be appropriate for publication in the journal provided that the authors can address the following points.
Minor:
#1: Authors should describe the reason why RJ treatment can improve serum LH and FSH levels. Have you checked histopathological changes in pituitary gland of the animals?
#2: Authors should delete primer sequences of Bcl2, Bax, and Caspase 3 in Table 1 because I can not find the RT-PCR data of these genes in this paper.
#3: Page 5, line 138. ...GPx, GR, SOD,... --> ..GPx1, GR, SOD2,...
Author Response
#1: Authors should describe the reason why RJ treatment can improve serum LH and FSH levels. Have you checked histopathological changes in pituitary gland of the animals?
Response: Thanks for your comments, no we haven’t study the histopathological alterations in the pituitary gland to understand the effect of RJ on sex hormones, but we will take this point in our considerations in future experiments.
#2: Authors should delete primer sequences of Bcl2, Bax, and Caspase 3 in Table 1 because I can not find the RT-PCR data of these genes in this paper.
Response: We followed your comment and the mentioned primers have been deleted.
#3: Page 5, line 138. ...GPx, GR, SOD,... --> ..GPx1, GR, SOD2,...
Response: We followed your comment.
Reviewer 2 Report
The present study examined the efficacy of royal jelly (RJ) against cadmium chloride (CdCl2)-induced testicular dysfunction in 28 male Swiss mice.
Introduction provides sufficient background and includes relevant references. The research design is appropriate and the conclusions are enough reported by the results.
My recommendation is to accept after minor revisions related to
Moderate English changes required
Line 106: RJ pre-treatment had a significantly suppressed the decrease in the testes weight 106 caused by CdCl2 injection.
Line 186: many histopathological alternations: degenerative changes
Line 220: This weight decrease may be due a decrease in
Figures
Figure 9. Photomicrographs showing changes in caspase-3, Bax, Bcl-2, and PCNA expression…
I suggest changing the order and using the same of the photomicrographs Bcl-2, Bax, Caspase 3 and PCNA expression….
Discussion
How did the authors explain LH and FSH increases in RJ groups? Did you measure activin or inhibin? Or another marker related to Sertoli cells?
The authors say that the decrease in the absolute and relative testicular weight may be due to a decrease in food intake. Have you measure food intake?
Line 251-252: According to our previous study, the increased nitrate/nitrite level is due to iNOS upregulation. Did the authors measure nNOS and iNOS mRNA expression or activity in testicular tissue?
References
In general, I suggest citing the name of the authors instead of the reference number when specifying data of this article in particular.
Line 230: [34] explained the increase of testosterone…
Line 259: [45] attributed CAT deactivation to the Fe deficiency..
Line 305: [60] demonstrated that RJ consumption….
Author Response
Moderate English changes required
Line 106: RJ pre-treatment had a significantly suppressed the decrease in the testes weight 106 caused by CdCl2 injection.
Line 186: many histopathological alternations: degenerative changes
Line 220: This weight decrease may be due a decrease in
Response: Thanks for your comments and we followed your suggestions and all mentioned mistakes have been corrected.
Figures
Figure 9. Photomicrographs showing changes in caspase-3, Bax, Bcl-2, and PCNA expression…
I suggest changing the order and using the same of the photomicrographs Bcl-2, Bax, Caspase 3 and PCNA expression….
Response: We followed your comment.
Discussion
How did the authors explain LH and FSH increases in RJ groups? Did you measure activin or inhibin? Or another marker related to Sertoli cells?
Response: We have measured only the concentration of LH and FSH in serum.
The authors say that the decrease in the absolute and relative testicular weight may be due to a decrease in food intake. Have you measure food intake?
Response: We have used this explanation according to previous studies, which have been mentioned in this part.
Line 251-252: According to our previous study, the increased nitrate/nitrite level is due to iNOS upregulation. Did the authors measure nNOS and iNOS mRNA expression or activity in testicular tissue?
Response: We have measured previously the expression of iNOS in liver and brain tissues.
References
In general, I suggest citing the name of the authors instead of the reference number when specifying data of this article in particular.
Line 230: [34] explained the increase of testosterone…
Line 259: [45] attributed CAT deactivation to the Fe deficiency..
Line 305: [60] demonstrated that RJ consumption….
Response: We have followed your suggestion.